# Image Filtering to Improve Maize Tassel Detection Accuracy Using Machine Learning Algorithms

**DOI:** 10.3390/s24072172

**Published:** 2024-03-28

**Authors:** Eric Rodene, Gayara Demini Fernando, Ved Piyush, Yufeng Ge, James C. Schnable, Souparno Ghosh, Jinliang Yang

**Affiliations:** 1Department of Agronomy and Horticulture, University of Nebraska-Lincoln, Lincoln, NE 68583, USA; eric.rodene@huskers.unl.edu (E.R.); schnable@unl.edu (J.C.S.); 2Center for Plant Science Innovation, University of Nebraska-Lincoln, Lincoln, NE 68583, USA; 3Department of Statistics, University of Nebraska-Lincoln, Lincoln, NE 68583, USA; 4Department of Biological Systems Engineering, University of Nebraska-Lincoln, Lincoln, NE 68583, USA

**Keywords:** UAV imagery, high-throughput phenotyping, machine learning, convolutional neural network, object detection, maize tassel detection, image segmentation

## Abstract

Unmanned aerial vehicle (UAV)-based imagery has become widely used to collect time-series agronomic data, which are then incorporated into plant breeding programs to enhance crop improvements. To make efficient analysis possible, in this study, by leveraging an aerial photography dataset for a field trial of 233 different inbred lines from the maize diversity panel, we developed machine learning methods for obtaining automated tassel counts at the plot level. We employed both an object-based counting-by-detection (CBD) approach and a density-based counting-by-regression (CBR) approach. Using an image segmentation method that removes most of the pixels not associated with the plant tassels, the results showed a dramatic improvement in the accuracy of object-based (CBD) detection, with the cross-validation prediction accuracy (*r*^2^) peaking at 0.7033 on a detector trained with images with a filter threshold of 90. The CBR approach showed the greatest accuracy when using unfiltered images, with a mean absolute error (MAE) of 7.99. However, when using bootstrapping, images filtered at a threshold of 90 showed a slightly better MAE (8.65) than the unfiltered images (8.90). These methods will allow for accurate estimates of flowering-related traits and help to make breeding decisions for crop improvement.

## 1. Introduction

One of the ultimate goals of plant breeding is to enhance traits such as disease resistance, yield, and stress tolerance, as well as water and nitrogen use efficiency. High-throughput phenotyping (HTP) is an essential tool in furthering crop improvement efforts and refers to the use of advanced sensing, robotics, and automation to collect plant phenotypic data cost-effectively and across multiple scales [1]. In-field HTP for crop improvement has been of great importance as it allows large numbers of plants in field trials to be analyzed quickly and, therefore, has the potential to increase the selection response per unit of time and contribute to global food security. The integration of in-field HTP into genomic selection protocols may further improve the selection efficiency, especially for crop traits under complex genetic control [2].

The maize tassel, composed of male flowers, is one of the highly complex traits and the major reproductive structure of the maize plant [3]. The timing of tassel emergence is critical for maize reproduction and local adaptation and is one of the most essential yield component traits. Conventional methods to manually measure the male flowering time traits, such as days to anthesis, by walking the rows of the test plots to record manual observations of tassel counts per row, are both tedious and time-consuming. Therefore, automated methods of both detecting and counting maize tassels (or sorghum panicles) are attractive, as they enable detailed quantification of traits related to flowering time, i.e., panicle counts in sorghum [4] and the duration of flowering time for a given genotype. The latter trait will contribute to making more valuable crossing decisions from pollen donor to pollen receiver, as some of the female and male organs might not be ready at the same time.

Machine learning is a subset of artificial intelligence and includes deep learning and convolutional neural networks (CNNs). CNNs are one common type of architecture used in deep learning, and have been applied specifically to computer vision tasks, such as object detection, image segmentation, and image classification [5]. These techniques have seen a variety of uses in agronomy- and horticulture-related studies. They have been used to detect seedling emergence in lab-grown experiments, as well as to determine plant germination rates [6]. They can be used to detect and count fruit in images of trees [7]. Several studies have made use of these methods for disease detection in various plant species [8,9,10]. Machine learning has also been applied to estimate plant density and leaf area [11,12,13]. Computer vision methods have also been applied to automated robots for picking fruit [14,15,16] and have also been applied to pathfinding algorithms and to enable automated obstacle avoidance [15]. It has also been found that CNNs trained on synthetic data or real image data supplemented with synthetic images are a viable approach to crop leaf counting. This synthesized image data can be beneficial in cases where high-quality training datasets using real images are limited or unavailable [17].

Recently, several machine learning algorithms, such as ResNet [18] and YOLO [19], have been developed for tassel detection from unmanned aerial vehicle (UAV)-based photography [20,21,22,23,24,25]. The algorithms can be characterized into two major categories: counting by detection (CBD) and counting by regression (CBR) [26]. In the CBD approach, such as Faster R-CNN [27], RetinaNet [28], and ResNet [18], the number of tassel objects in a field plot can be measured by first identifying the objects and subsequently counting them. In the CBR approach, such as TasselNet [20] and YOLO [19], the number of tassels can be computed directly from the image using regression models without object detection [29]. The existing algorithms have mainly been developed for a limited number of genotypes in large-scale field trials. Applying these algorithms for small-plot field trials with hundreds of different genotypes is challenging, as typical small-plot field images tend to contain many complex effects, such as the diversity of tassel morphology, a difference in plant architecture and biomass, and hence a distinct interaction of light and shadows on the plant leaves under different lighting conditions.

In this study, by leveraging our previously published UAV imagery data collected in a maize field trial for 233 different genotypes [30], we developed a two-step tassel detection method to first filter the images to remove noisy elements and then applied two different machine learning algorithms, counting-by-detection (CBD) and counting-by-regression (CBR), to the filtered images for tassel detection. The two-step results implemented in TasselNet, a CBR algorithm, did not outperform the conventional non-filtering approach. However, as compared with non-filtering approaches, the CBD approach chosen (see Materials and Methods, Section 2.4) showed a considerable improvement in the detection accuracy of individual tassels, and in particular accuracy of tassel counts, compared to ground truth data. The two-step tassel detection approach developed in this study, particularly designed for small-plot field trials with diverse maize genotypes, can be integrated into plant breeding by using the male flowering time traits as a proxy for reproduction and yield traits and can be used for making crossing decisions.

## 2. Materials and Methods

### 2.1. Field Experimental Design and Image Data Collection

The images used in this study are UAV-based aerial photographs taken during the summer of 2019 on a maize diversity panel [30]. The test plots were grown at the Havelock Research Farm at the University of Nebraska-Lincoln using an incomplete split plot block design. The field was arranged into four quadrants (NE, NW, SE, and SW). The NE and SW quadrants were nitrogen-treated (+N), i.e., treated with 120 lbs/acre (approximately 134.5 kg/ha) of urea. The NW and SE quadrants received no nitrogen treatment (−N). The quadrants were each divided into six ranges, with 42 two-row plots per range, each plot representing a different genotype. The rows were spaced 30 inches (76.2 cm) apart. The within-row spacing was 6 inches (15.24 cm). Each two-row plot was 5 ft × 20 ft (1.524 m × 6.096 m).

In total, over 112,000 plot-level images were obtained in a time-series manner for 233 different genotypes planted under high nitrogen (N) and low N conditions. As stated in [30], the UAV images were collected in a series of 12 flights from July 6 to September 5 using a Phantom 4 Pro UAV equipped with an RGB camera. Each image has a resolution of 5472 × 3648 pixels (see CyVerse dataset 10.25739/4t1v-ab64 for data availability). The software Plot Phenix was used to extract the plot-level images. These extracted images had variable resolutions, but were typically between 250 × 1000 and 300 × 1200 pixels. A selection of these plot-level images was used as the basis for this study’s training and testing dataset. A total of 323 images were selected based on the plot’s quadrant (NE, SE, NW, or SW), growth tendency (tall or short), flowering time (early or late), and nitrogen treatment (+N or −N). The same genotypes were selected for each date that UAV data had been collected where tassels were visible in the plot (see Appendix A). This dataset of 323 images was randomly split up into one subset used for algorithm training and a second subset used to subsequently test the prediction accuracy of the trained algorithms, as compared to the ground-truth tassel counts.

### 2.2. Procedure for Image Filtration

Image filtration was accomplished in two parts. The first part was to remove non-foliage elements using the Excess Green Index (ExG) [31]:(1)$$2G^{*}-R^{*}-B^{*}$$

Here, the values of $G^{*}$, $R^{*}$, and $B^{*}$ are calculated based on the green, red, and blue channels, respectively, of each pixel of the image and dividing this value by 255, the maximum value possible in the RGB color spectrum for 8-bit integers. The calculated ExG values were further modified by rescaling them to range from 0 to 255 to make threshold selection simpler than it would be when dealing with decimal values. Note that this normalization also differs from what is presented in Woebbecke’s paper, though, as we explain in our earlier paper, this normalization was found to perform more accurately for image segmentation. Previous work conducted using this same image dataset has indicated a threshold of 131 provided the most promising results for filtering of non-foliage elements for the image dataset used [30]. It should also be noted that this filtering method does not result in significant elimination of the tassels at the threshold chosen for this image dataset (see Appendix A, which leverages the same pixel sampling method described with Figure 1). At a threshold of 131, our pixel sampling tests indicated the ExG removed over 99% of the sampled soil pixels and over 97% of the sampled shadow pixels, but only 0.5% of the sampled foliage and 13.5% of the sampled tassel pixels. This generally ensured that the structure of the tassels was still sufficiently easily visible in the filtered images.

Once the images were filtered by the ExG, the second part of the filtering process was to remove the foliage, leaving behind only the tassel pixels. This was accomplished by using a second formula:(2)$${\left[2\left(G^{*}+R^{*}\right)\right]}^{2}-B^{*}$$

As described with the ExG (Equation (1)), the resulting value was then rescaled to range from 0 to 255. This formula was selected out of a variety of other possibilities devised by our in-house tests, which indicated it had the best performance in distinguishing between tassels and foliage. The rationale behind the structure of the formula was that because the ExG is designed to assign high values to green pixels [31], in order to filter tassels, a new formula would need to assign higher values to yellowish pixels, as the tassels tend to be more yellowish compared to the foliage. In the RGB color spectrum, pixels with high green and red values but low blue values tend to be yellowish to the human eye. Because this formula combines the red and green values, doubles the result, and then additionally squares it, this provides high scores to pixels with high green and red values but low blue values, while generally assigning lower scores to pixels with other combinations of RGB values.

The assessment of the above proposed formula was accomplished by applying it to a set of sampled images. A series of 150 plot-level images were marked for examples of soil, shadows, foliage, and tassels (Figure 1a). These images were then iterated using thresholds ranging from 0 to 255. Anytime a marked pixel was encountered, the formula was then applied to the corresponding pixel in the unmarked version of the same image. Whenever the calculated value was less than the threshold of the current iteration, it was eliminated. The results of this analysis (Figure 1b) were used to (a) determine if Equation (2) was suitable for distinguishing the different labeled features of crop images, and (b) to determine an approximate threshold which would generally produce low elimination of the tassel pixels while eliminating most of the other features. An ideal filter should eliminate 100% of the non-tassel pixels, while leaving all pixels associated with the tassels unaffected.

Equation (2) was also compared with the Excess Red Index (ExR) [32], which has the following formula:(3)$$1.4R^{*}-G^{*}$$

As before, the values of $R^{*}$ and $G^{*}$ are produced by dividing the red and green channels of the pixel by 255, and the calculated ExR value was further rescaled to range from 0 to 255. The results of a recent study suggest that the ExR shows a high sensitivity to the presence of tassels in aerial maize images [33], and so this index was deemed worthwhile to investigate further in this regard.

Once again, using our proposed formula (Equation (2)), the scores from the second filtering step were re-scaled to range from 0 to 255. It was found that the ideal filter threshold varies substantially from image to image. However, this can be selected using the cross-validation steps. Specifically, graphs were generated giving the *r*^2^ values for the selection of test images, comparing the number of tassels predicted by the detector with the ground-truth observation. This resulted in a range of possible threshold values which tended to generate the highest *r*^2^ values, generally between the thresholds of 90 and 110 (see Section 3.3 and Figure 4).

### 2.3. The TasselNet Approach

We used the original TasselNet architecture and first trained it on the Maize Tassel Counting dataset [20]. The parameter estimates obtained from this training round of TasselNet were used to initialize the deep learner in our data. In the final full-scale deployment of TasselNet, we used a random sample of 258 plot-level images for training purposes, tested the predictive performance on the remaining 65 images, and performed a five-fold cross-validation. Each image was subdivided into 32 × 32 sub-images (with the stride length set at 8), creating a 32 × 32 × 3 tensor. The array of tensors obtained by concatenating the sub-images formed the basic input block in the CNN. A set of 269,422 tensors was randomly selected as a validation set to enforce early stopping. We used four convolutional layers that corresponded to the LeNet architecture of TasselNet [20,34] and invoked dropout layers and batch normalization to reduce over-fitting. The model was trained for 50 epochs with a patience of 5. The learning rate and bandwidth of the Gaussian kernel are two key hyperparameters. We used validation loss to optimize these two hyperparameters. A learning rate of 0.0001 and a bandwidth of 8 turned out to be optimal in our situation. We experimented with several popular pre-trained networks (for example, VGG16 and VGG19) for feature extraction purposes and then processed the local count regression via the TasselNet framework. We observed that using pre-trained networks (for feature extraction in tassel images) produced inferior out-of-sample predictions as compared to the deep learner exclusively trained on our own dataset. Hence, we report the mean absolute error (MAE) for the latter case only. In order to obtain bootstrap-based uncertainty intervals for the MAEs, we used an algorithm that was similar to the one outlined in [35].

Because TasselNet requires training on overlapping sub-images, it becomes computationally expensive to train this model even for moderate sample sizes. For instance, 258 images yielded a total of 1000 K sub-images and a full-scale deployment required approximately 15 h per image set. Therefore, to reduce the computation time, we assessed the performance of TasselNet trained on a smaller training set. The smallest training set that we considered was a random sample of 250 K sub-images. A total of 100 bootstrap replicates (of size 250 K) of sub-images were drawn from the full set of training sub-images, and we generated predictions on a fixed holdout test set.

The technical details of TasselNet are discussed in [20]. For the sake of completeness, we summarize the technical aspects of this method here. TasselNet first obtains a density map by smoothing the point-referenced image that identifies the locations of the tassel in the original image. A bivariate Gaussian smoother of a fixed bandwidth is used to generate the density map (see Appendix A). TasselNet then decomposes each density map (I) into a set of overlapping sub-images of density maps ($I_{s}$). These sub-images are generated using a stride length chosen a priori. If ${D(s}_{i})$ denotes the density at a location $s_{i}$, then the regression target of TasselNet for the jth sub-image ($I_{s}^{j}$) is given by $T^{j}=\sum_{s_{i}\in I_{s}^{j}}D(s_{i})$. Observe that the regression target of TasselNet can be viewed as an approximation of the expected count in each sub-image. In the current construction, TasselNet uses an $L_{1}$ loss function given by $L_{1}=\frac{1}{M_{tr}}\sum_{j=1}^{M_{tr}}\left|T^{j}-\hat{T^{j}}\right|$, where $M_{tr}$ is the total number of training sub-images and $\hat{T^{j}}$ is the estimated count produced by the chosen convolutional neural network. During the prediction phase, each image is scanned by a sliding window of a specified size. Predicted counts are generated for each sub-image induced by the sliding window. The predicted count from each sub-image is first uniformly redistributed to each pixel in the sub-image and the number of times each pixel is counted is tracked. The final predicted count of a test image is given by $\sum_{x,y}\frac{C(x,y)}{P(x,y)}$, where C(x,y) is the redistributed count assigned to the pixel (x,y) and P(x,y) is the number of times pixel $\left(x,y\right)$ was counted.

Since the regression targets are counts and an “absolute error” loss function is used, the MAE turns out to be the most suitable to assess the performance of TasselNet. Observe that Tasselnet is not a classifier. It targets a non-zero scalar quantity for each input sub-image. Consequently, accuracy and recall measures are not available for this method. We therefore produce two additional performance metrics that are suitable for count regression:(a)Normalized mean square prediction error (NMSPE).

We compute this quantity in the following way:

Let $T_{k}$ be the observed number of tassels in a test image, and let $\hat{T_{k}}$ be the predicted count produced by Tasselnet. Then, we define the NMSPE metric as:(4)$$\frac{1}{M_{ts}}\sum_{k=1}^{M_{ts}}\frac{{\left(T_{k}-\hat{T_{k}}\right)}^{2}}{\hat{T_{k}}}$$

Observe that this metric is similar to the Chi-squared statistic typically used to assess goodness-of-fit in count data.

(b) Predictive deviance (PD).

The proposed PD metric is inspired by the deviance measure typically used to assess goodness-of-fit in Poisson regression. It is defined as:(5)$$2\sum_{k=1}^{M_{ts}}\left[T_{k}\log\left(\frac{T_{k}}{\hat{\;T_{k}}}\right)-\left(T_{k}-\hat{T_{k}}\right)\right]$$

If $T_{k}$ is 0, then the leading term is taken to be 0. Smaller values of NMSPE and PD indicate a higher predictive accuracy.

Table 1 above shows the foregoing NMSPE and PD measures under five-fold cross-validation. Both these metrics suggest TasselNet performed relatively better with unfiltered images. The results are in agreement with the cross-validated MAE reported earlier.

In addition, since $T_{k}$ is a count variable in the test set and $\hat{T_{k}}$ is continuous in nature, we report the Spearman’s correlation between $T_{k}$ and $\hat{T_{k}}$ across the folds in the following table.

Once again, the results (see Table 2) suggest the relatively superior performance of TasselNet on unfiltered images.

### 2.4. The R-CNN Approach

The machine learning process was conducted using the “trainRCNNObjectDetector” function in Matlab on a detector pre-trained using the CIFAR-10 image dataset [36]. This technique uses an R-CNN object detector, which relies on a CNN to classify regions in a given image [37,38]. The R-CNN is designed to process only the region proposals likely to contain the objects being detected. This implementation relies on a transfer learning workflow, where the network is initially trained on a large image dataset (CIFAR-10 in this case). The learning can be transferred to a new object-detection procedure by fine-tuning the network by adjusting the feature representations learned in the pre-training step to support the new task. This allows the training dataset (in our case the tassel images) to not need to be so large, reducing required training times. In this training step, the input network weights were adjusted using sub-images extracted from the ground-truth data annotations in the training image set. Model training was conducted over 100 epochs, with a mini batch size of 128, an initial learning rate of 0.001, a negative overlap range from 0 to 0.3, and a positive overlap range from 0.5 to 1. The parameters used were mostly default values. There is also a tradeoff with parameters such as the number of epochs and mini batch size. More epochs will more thoroughly train the detector to recognize tassels, but this also means more time is needed for the training step to complete, and a larger mini batch size will mean a more demanding memory usage.

The set of 323 annotated plot-level images was used to conduct randomized sample testing to evaluate the model performance. In this study, instead of splitting the image set into two equal subsets for training and testing, 80% of the image dataset (258 images) was randomly selected for the training set after being filtered at various thresholds for different test runs (90, 95, 100), while the remaining 20% (65 images) was used for the testing step, during which the trained detector was applied. The test images were iteratively filtered at thresholds from 0 to 255. The predicted tassel counts at each threshold were then collected and compared to the ground truth data to calculate the prediction accuracy using correlation coefficients (i.e., *r*^2^ values). This allowed the *r*^2^ curves to be plotted against the filter threshold of the test images to see what thresholds appeared to have the highest *r*^2^ values in that selection. This assessment also differs from the MAE score used to evaluate TasselNet’s accuracy, as it uses a density-based approach to identify tassels; this fundamentally differs from the CBD approach employed here. For the CBD approach, we are most interested in how closely the predicted tassel counts match the ground-truth data; therefore, the *r*^2^ metric was used in this case. One main advantage of the CBD method over TasselNet’s CBR approach is less computational time. The use of transfer learning from the CIFAR-10 dataset allows the training dataset of tassel images to be smaller, which leads to shorter training times. The CBD method also only focuses on region proposals with a high likelihood of matching the patterns of the training data, further decreasing the computational time compared with the CBR approach previously discussed.

## 3. Results

### 3.1. Image Filtration Removes Non-Tassel Pixels

In our experimental design, 233 different maize genotypes, representing the extensive genetic diversity of modern maize [39], were grown in a field split into two replicates of nitrogen-treated (+N) quadrants, and two replicates of non-treated (−N) quadrants [40]. Images were collected via a UAV throughout the growing season under various different imaging conditions and were used as the basis for this study [30]. The raw images were stitched using the orthomosaic software Phenix, and individual plots were clipped from these images for further analysis. Furthermore, a two-step filtering procedure was devised to remove both background and foliage, leaving behind pixels associated with tassels (see Section 2. This filtering approach is similar in concept to that described by Zhao et al. used for panicle counting in sorghum [41] but follows a different method to achieve it. We have tested our approaches using both the filtered and unfiltered images. An overview of the image collection and filtering pipeline is shown in Appendix A.

From the plot-level images collected, a representative subset of 323 plot-level images was selected and manually annotated with bounding boxes for each tassel to serve as training data, as well as to provide ground-truth data for testing purposes (see Appendix A). Images were selected from different dates from both nitrogen treatments and for different developmental stages of the diversity panel (see Section 2).

Both the Excess Red Index (ExR) [32] and the tassel filtering formula we have devised here (Equation (2)) were found to produce similar results on most plot images after the images were pre-filtered using the Excess Green Index (ExG) [31] (see Section 2). However, the elimination results of both indices differ markedly (Figure 1b and Appendix A). Furthermore, it was found that the ExR performs poorly at distinguishing between soil and tassels when unfiltered images are used, while the tassel filtering formula still adequately removes most non-tassel pixels (Appendix A). The main challenge of our proposed formula is it can have difficulty distinguishing between tassels and debris in unfiltered images (Appendix A). Because both formulas produce similar results in filtering the test images, and because our proposed formula had already been used to filter images which were then used to train the detector for a series of early five-fold cross-validation steps, it was not deemed worthwhile to repeat the training and validation steps using the ExR, as the results of doing so can be expected to be very similar to what has already been observed.

### 3.2. Filtered Image Feeds into a CBR Algorithm Implemented in TasselNet

Machine learning algorithms have been applied to tasks such as panicle detection in sorghum [4] and tassel detection in maize [22,23]. To generate tassel count estimates for each of our plot-level images and to then track how the counts for each individual plot change as the plants develop to maturity, we investigated the accuracy of both a CBR (TasselNet) and a CBD approach (see Section 2), following the procedure as illustrated with the R-CNN used for the CBD approach (Figure 2).

For the CBR approach, TasselNet was deployed on the unfiltered and filtered images separately (see Section 2). This protocol yielded a cross-validated mean absolute error (MAE) of 7.99 for unfiltered images (see a visual representation of the tassel localizing ability of TasselNet in Figure 3). The cross-validated MAEs using filtered images with thresholds set at 90 and 100 turned out to be 8.59 and 10.39, respectively. Numerically, it appeared that the predictive performance of TasselNet was relatively better when unfiltered images were used as inputs.

The bootstrapped mean MAEs for the test set, along with a naïve 95% bootstrapped uncertainty interval for these MAEs, are shown in Table 3. As expected, the predictive performance of TasselNet did suffer as the training set became smaller—with the decline being more pronounced in the case of unfiltered images as compared to filtered ones. Although the uncertainty intervals suggested that the MAEs for unfiltered and filtered images (with threshold set at 90) were statistically similar (see Table 3), the predictive performance of TasselNet trained on images with the filtering threshold set at 100 was statistically inferior. Regardless, the results suggested that filtering was beneficial when dealing with a smaller sample size.

### 3.3. Filtered Images Improved Tassel Detection for a CBD Algorithm

In contrast to the CBR approach, the CBD approach (see [18,27,28] for some example algorithms) works by detecting patterns in input images similar to those they have been trained to recognize. These region proposals will be marked with bounding boxes, indicating that the algorithm has detected a pattern in that region with a high-scoring match to the training data. This approach has been used for object-based detection of tassels [22,23] and sorghum panicles [4].

The CBD algorithm used here typically finished training within 19 h using 258 images. The filtering of the training images is a separate matter from the filtering of the images the detector is to be tested on. Early analysis indicated that filter thresholds between 90 and 110 generally produced the best results for the training images (Figure 4). The test images were instead iteratively filtered to varying thresholds from 0 to 255 (see Section 2). The highest *r*^2^ value observed was 0.7033 when training with images filtered at a threshold of 90 (Figure 4). This was also noticeably higher than the *r*^2^ values observed using the unfiltered images (0.4162 being the highest value observed). Also note that there was a similar spike in *r*^2^ values when the test images were filtered to thresholds of around 90, even when the detector used had been trained on unfiltered images. This result suggests that the *r*^2^ spikes are likely not due to the filter thresholds somehow biasing the results when the test image filtering threshold approaches the same threshold used for the training images. If there was a bias, it would be expected that the detector trained with unfiltered images would instead see a spike at very low thresholds where little filtering has taken place (the *r*^2^ value when the version of the detector trained on unfiltered images was also applied to completely unfiltered versions of the test images was only 0.0761). Instead, we believe that filtering the test images from our dataset to thresholds around 90 tends to eliminate most of the noisy elements from extraneous pixels which otherwise seem to confuse the accuracy of this type of detector.

Leveraging the set of 323 images produced much clearer results compared to initial tests, and also confirmed that filtering of the images at thresholds between 90 and 110 generally produces the most accurate results compared to the ground-truth data (Figure 4).

## 4. Discussion

This study represents an effort to improve the tassel detection accuracy of plot-level images for diverse maize genotypes by using a customized filtering approach to remove most of the non-tassel pixels. Using the CBD methods available in Matlab (based on those described in [37,38]), our results showed that this approach dramatically improves the accuracy of tassel detection, both by aligning the predicted numbers of tassels more closely with the ground-truth data and also with the placement of the bounding boxes over region proposals showing a high probability of containing tassels. This is not surprising, as the filtering methods we have employed have the effect of removing extraneous pixels, which undoubtedly contribute unnecessary noise to the detection process, confounding the detection accuracy when using unfiltered images. It is anticipated that this image segmentation would show similar results in other CBD methods.

Using the plot-level images for diverse maize genotypes, our tests using the annotated database of 323 images indicate some of the best object-based detection results when the training images are pre-filtered to a threshold of 90. Likewise, we observed the highest detection accuracy when applying the trained detector to test images filtered to thresholds between 90 and 100 (Figure 4b). In contrast, the CBR methods employed by TasselNet produced the lowest MAE values when using the unfiltered training images. As this skips the two-part filtering step, in some ways, this method may be simpler to use, although its main weakness is in its scalability, as larger training datasets will take considerably more computational time to complete. Regardless, TasselNet is equipped with inferential capabilities in the sense that the local regression formulation essentially produces an intensity map from a spatial point pattern input, with the point pattern annotating the location of each tassel in the image. Subjecting the point pattern to Gaussian smoothing produces an intensity map. The CNN-based regression function models this smoothed random field associated with the intensity map of tassel occurrence. Evidently, integrating the output intensity map (see Figure 3, left panel) over the bounded domain of the image will produce an expected tassel count in that image. Now, a formal likelihood-based inference using techniques developed for spatial point processes [42] could be used to rigorously assess the uncertainty associated with the estimates produced by TasselNet. In fact, we argue that the CBR and CBD techniques discussed herein complement each other and could be integrated to rigorously assess the uncertainty in tassel count estimates produced via CBD techniques. A straightforward way to perform such integration is to use CBD techniques to estimate the direct counts and predict the occurrences of tassels in the input images (see Figure 4a). TasselNet can take this point pattern of tassel occurrence and predict an intensity map, which can be subsequently processed using a likelihood-based technique to quantify the uncertainty in the counts generated using the CBD technique. This integrated model could potentially be extended to perform model selection or model averaging.

The scalability challenge for the TasselNet approach arises from the fact that it trains over multiple overlapping sub-images. Consequently, even for our relatively small sample size of 258 full images, TasselNet was trained on 1000 K sub-images of size 32 × 32. This augmentation procedure, hard-wired into TasselNet, is the principal computational bottleneck for this method. We encountered this computational constraint when we deployed bootstrapping and ended up using a quarter of the original 1000 K training sub images. Our preliminary investigation suggests that if we deploy the original version of TasselNet on a CPU-based high-performance computing cluster on a training set consisting of a few thousand images, the training time would exceed 24 h. A computational time on the order of days limits the practical utility of this method. We therefore recommend training TasselNet on a smaller sub-sample of sub-images. The number of sub-images and the size of each sub-image are two hyperparameters that will determine the computational time. Our investigation suggests that with 32 × 32-sized sub-images, a training set on the order of 10^5^ sub-images can be reasonably deployed on a high-performance computing cluster. Another potential path to make TasselNet more scalable is to deploy it on a GPU-enabled high-performance computing cluster.

The tassel filtering and machine learning approaches we have detailed do have some drawbacks, however. The pipeline requires manual image annotation for the training data if the dataset does not already have that information available, which means that producing a reasonably sized training dataset can be quite time-consuming. The ExG and tassel filters necessitate the pixel sampling discussed in Section 2.2 and in Figure 1 to determine elimination rates for the filters, as well as to select a reasonable threshold for the dataset in question, as it is expected that the threshold selected for one image dataset may not necessarily be the optimal choice with a different dataset owing to possible differences in the lighting, the quality of the images, and the camera settings used to collect the images. There is also the drawback that picking a single static threshold may produce better filtering results with some images within the dataset than others. A future improvement that could be anticipated is to address the need for a more dynamic filtering approach, instead of expecting a one-size-fits-all solution to work equally well with all images. Our study also only focused on filtering the images by blacking out pixels flagged for elimination and leaving the other pixels as they are. It was not investigated how it would affect the detection performance if the surviving pixels were instead used to create a simple binary mask for the training and testing images. This could also be studied in future work.

The tassel detection methods outlined here allow users to produce plot-level, time-series tassel counts. As flowering-time-related traits are correlated with plant fitness and grain yield, the results generated by applying our methods for UAV data analysis can provide valuable information for future plant-breeding efforts.

## Figures and Tables

**Figure 1 sensors-24-02172-f001:**
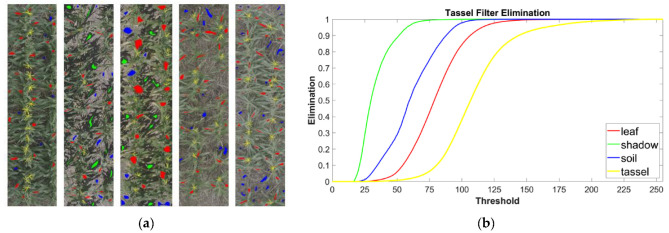
**Overview of the tassel filter and sampling process.** (**a**) Five image examples of the sampled two-row plots. Images were marked for soil (blue), shadows (green), foliage (red), and tassels (yellow). (**b**) The elimination results of tassel filtering. This shows which elements (soil, shadows, foliage, and tassels) are eliminated by thresholding the calculated score of the sampled pixels. As the threshold increases, greater percentages of the sampled pixels are eliminated, until ultimately, 100% elimination of all sampled features is reached at the highest thresholds.

**Figure 2 sensors-24-02172-f002:**
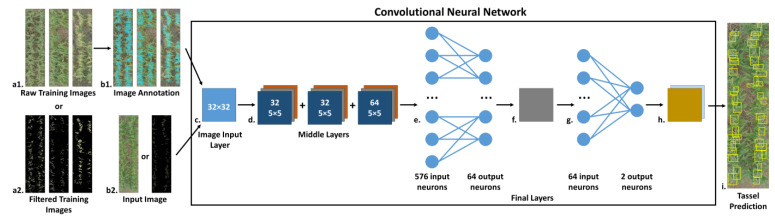
**A basic overview of the tassel prediction pipeline in the CBD approach.** A training dataset of plot-level images (**a1**) is selected and manually annotated to mark locations of tassels visible in the images with bounding boxes (**b1**). Alternatively, these same images can also be filtered to remove pixels associated with non-tassel elements (**a2**). The annotation data remain the same regardless of image filtering. The training dataset and annotation coordinates are then used to train the CNN. Once trained, input images (**b2**) can be fed to the CNN for tassel prediction. As with the training images, the input images may alternatively be filtered to eliminate extraneous pixels that can reduce the prediction accuracy. The CNN itself consists of a 32 × 32 image input layer (**c**); a series of three clustered middle layers, each consisting of a convolutional layer with 32 5 × 5 convolutions (or 64 convolutions in the final cluster); a rectified linear unit; a 3 × 3 max pooling layer (**d**); and a final set of layers consisting of one fully connected layer with 64 output neurons (**e**), a rectified linear unit (**f**), an additional fully connected layer with two output neurons (one to identify tassels and the other to identify background) (**g**), a SoftMax layer, and a final classification layer (**h**). The predicted tassel regions are then marked with bounding boxes in the input image (**i**).

**Figure 3 sensors-24-02172-f003:**
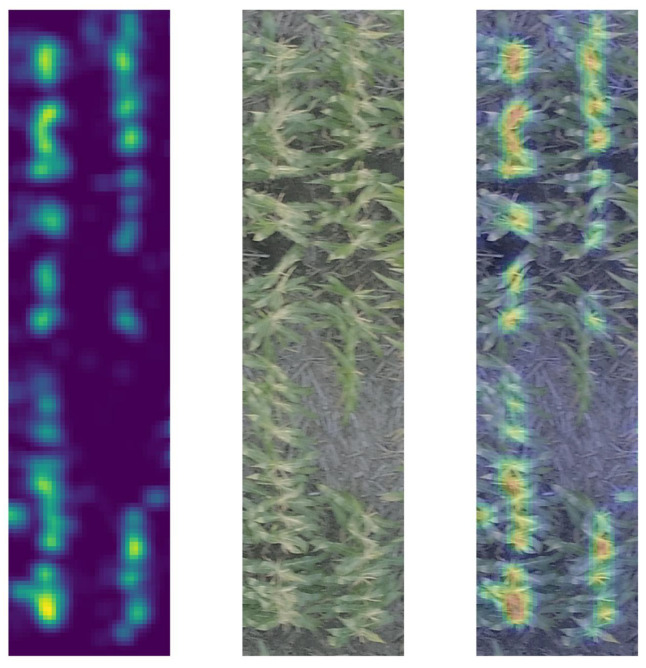
**Heat maps for a test case.** In the left panel is the heat map, in the middle panel is the original unfiltered image of the test case, and in the right panel is the heatmap superimposed on the original image of the test case. Warm-colored pixels indicate that TasselNet has detected the presence of tassels in the region covered by those pixels. Increase in the warmth of the color, from yellow to red, indicates that TasselNet predicts a higher density of tassels in the corresponding region. The cooler color indicates the sparsity of tassels as predicted by TasselNet. Blue pixels indicate that TasselNet has failed to detect the occurrence of tassels in the corresponding region.

**Figure 4 sensors-24-02172-f004:**
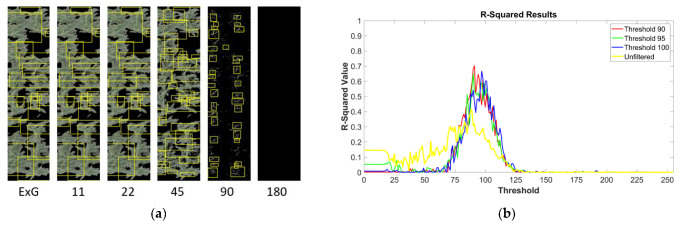
***r*^2^ values at varying tassel filter thresholds.** (**a**) The tassel filter at thresholds of 11 to 180 based on an image pre-filtered with the Excess Green Index (far left). Bounding boxes (yellow) indicate predicted tassels. (**b**) Average *r*^2^ values calculated based on the ground truth data. Each curve indicates the *r*^2^ values from using a randomized selection of images as the training data, filtered to thresholds of 90, 95, and 100. The yellow curve represents the results after training with the unfiltered images. The *X*-axis represents the threshold at which the sets of test images were filtered. The highest *r*^2^ value (0.7033) was observed at a test image threshold of 91 when training was conducted using images filtered to a threshold of 90.

**Table 1 sensors-24-02172-t001:** NMSPE and PD values for each image type.

Image Type	NMSPE	PD
Unfiltered	2.7231	180.8481
Filtered at threshold 90	3.8603	227.4901
Filtered at threshold 100	5.2032	312.1130

**Table 2 sensors-24-02172-t002:** Spearman’s correlation values for each image type.

Image Type	Spearman’s Correlation
Unfiltered	0.6825
Filtered at threshold 90	0.5802
Filtered at threshold 100	0.5041

**Table 3 sensors-24-02172-t003:** Average MAEs and their uncertainty intervals based on bootstrapped samples.

Image Type	Average MAE across Bootstrapped Samples	95% Uncertainty Intervals
Unfiltered	8.90	(8.73, 9.07)
Filtered at threshold 90	8.65	(8.58, 8.72)
Filtered at threshold 100	9.91	(9.83, 10.02)

## Data Availability

See Appendix A.

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
