# Peer review of "Image Filtering to Improve Maize Tassel Detection Accuracy Using Machine Learning Algorithms"

_sensors, 2024, doi:10.3390/s24072172_

Round 1

Reviewer 1 Report

Comments and Suggestions for Authors

In this manuscript, two techniques counting-by-detection (CBD) and counting-by-regression (CBR) based on machine learning were developed leveraging an aerial photography dataset taken by unmanned aerial vehicle (UAV). The authors described each part of the proposed method in detail, carried out extensive experiments and discussed the results seriously. The modification suggestions of this manuscript are given as follows.

1.     In rows 79-81 on Page 2 and rows 300-302 on Page 7, the authors say “The two-step results implemented in TasselNet, a CBR algorithm, didn’t outperform the conventional non-filtering approach.” and “ we investigated the accuracy of both a CBR (TasselNet) and ... the CBD approach (Figure 2).” Whether the TasselNet was used for CBR while the R-NN was used for CBD? In subsection 2.3 and 2.4, it is suggested to interpret the relationship between the network leveraged and the methods (CBD and CBR).

2.     In rows 138-141 on Page 3, the authors interpret the rationality of the structure of two-step filtration “The rationale behind the structure of the formula was that because the ExG is designed to provide high values to green pixels, in order to filter tassels, ..., as the tassels tend to be more yellowish compared to the foliage.” It seems that Equation (1) will filter the tassel, but Equation (2) tends to filter foliage while retaining tassel. Whether there is a contradiction because equation (1) is likely to filter out the tassel?

3.     In rows 135-138 on Page 3, it says “ This formula was selected out of ... between tassels and foliage.” It is suggested to add related references.

4.     It is suggested to add related references to “Specifically, graphs are generated giving the r^2 values for ... with the ground truth observation” in rows 172-174 on Page 4, or giving corresponding tables.

5.     What is the meaning of “LeNet” in row 187 on Page 4?

6.     The graphs and signals in Figure 2 are too small to display well. It is recommended to revise the figure and enlarge it to some extent. And CBD should be referred to if the network is used for it.

7.     It seems that there is only one dataset shown in Figure 4 to evaluate the method. More 1-2   other datasets should be added to confirm the effectiveness of the proposed method.

8.     There are some irrational sentences in the manuscript, listing as follows.

(1)In rows 29-30 of Page 1, “One of the ultimate goal of plant breeding is ..., as well as water and nitrogen use efficiency.” “goal”.

(2)In rows 43-45 of Page 1, “Conventional methods to ..., such as days to anthesis by walking ... and time-consuming.” A “,” is missing before “by walking”.

(3)In rows 46-47 of Page 2, “automated methods of both detecting and counting maize tassels (or sorghum panicles) becomes attractive” “becomes”.

(4)In rows 49-51 of Page 2, “The latter trait will contribute to making crossing decision from pollen donor to pollen receiver more precious as some of the female and male organs might not ready at the same time.” “crossing” and “ready”. Other crossing like one in row 86 on Page 2 should be checked and revised carefully.

(5)In rows 136-137 of Page 3, “ This formula was selected out ... it had the best performance with distinguishing between tassels and foliage.” “with”.

(6)In rows 360-362 from Page 8 to Page 9, “ The highest r^2 values (0.7033) were observed at ... a threshold of 90.” “were”.

(7)In row 396 on Page 9, a “with” is lacking after “associated”. In row 397, there is a “other” missing after “each”?

(8)In row 401-403 on Page 9, “TasselNet can take this point ... using likelihood-based technique to quantify the uncertainty in the counts generated by the CBD technique.” “technique”.

Comments on the Quality of English Language

 Minor editing of English language required

Reviewer 2 Report

Comments and Suggestions for Authors

The current manuscript presents a machine learning approach utilizing unmanned aerial vehicle (UAV) imagery for the counting of corn stalks. It employs object-based detection and counting (CBD) and density-based regression counting (CBR) to segregate non-tassel plant pixels through image segmentation. The paper reports that CBD significantly enhances accuracy in detectors trained on images. These methodologies are vital for accurately estimating flowering-related traits and inform breeding choices to advance crop improvement. Nevertheless, the document necessitates certain enhancements to align the reported results with the standards of scholarly publications.

General Considerations:

(1) It is suggested that Figure 3 be centrally positioned and replaced with a higher-resolution image.

(2) The paper references a total of 33 sources, comprising 15 from the past five years (46%), 11 from the last 5-10 years (33%), and 7 older than a decade (21%), culminating in 79% recent references. This proportion is insufficient; additional contemporary references are needed to strengthen the relevance and scholarly utility of the work.

Chapter 1: Introduction:

(3) The introduction effectively conveys the ultimate objectives of plant breeding, including disease resistance, yield, resilience, and water and nitrogen-use efficiency. It lays the foundation for the study and emphasizes the significance of drone imagery in corn ear detection. Refining the aforementioned aspects will enhance clarity and readability, facilitating the reader's transition to the methods and results sections.

(4) While the complexity of corn ears and the shortcomings of conventional measurement techniques are introduced, a comprehensive discussion of the specific machine learning methods to be applied is needed. More information on machine learning fundamentals would be beneficial.

(5) The introduction references the application of machine learning within agronomic and horticultural research yet does not directly relate this to the topic under investigation. The introduction should conclude with a clear tie-in to the remainder of the study. The authors may add more state-of-art computer vision articles in precision agriculture for the integrity of the manuscript (A Performance Analysis of a Litchi Picking Robot System for Actively Removing Obstructions, Using an Artificial Intelligence Algorithm; Agronomy. Path planning for mobile robots in unstructured orchard environments: An improved kinematically constrained bi-directional RRT approach; Computers and Electronics in Agriculture.Transforming unmanned pineapple picking with spatio-temporal convolutional neural networks. Computers and Electronics in Agriculture.).

Chapter 2: Materials and Methods:

(6) The manuscript clearly documents the field trial design and image data collection process, detailing test area conditions, fertilization, crop varieties, etc. Additionally, it describes the UAV image data, including resolution, equipment, and acquisition timing.

(7) While the experimental design, data sources, and processing stages are depicted adeptly, further clarification on the application of formulas and the TasselNet methodology is warranted. The description of TasselNet should include a more thorough evaluation of model performance metrics, such as accuracy and recall rates, to aid comprehension.

(8) The R-CNN method utilizes transfer learning from the CIFAR-10 dataset and optimization for the new task. More explanation is necessary regarding method detail, parameter choices, and a comparison with the TasselNet approach for reader comprehensibility.

Chapter 3: Results:

(9) This section details the image filtering process, datasets used, and CNN predictive modeling. To ensure reader understanding, it requires greater explanation of the comparative algorithmic performance and a more elaborate description of the CNN framework.

(10) A thorough evaluation of CBD methods has been undertaken, accounting for the influence of image filtering. The inclusion of quantitative metrics and visual aids heightens result clarity. The CBD method discussion should extend to explore why specific thresholding (90 to 110) yields accurate outcomes, offering deeper insights into algorithm functionality.

Chapter 4: Discussion:

(11) The discussion mentions the scalability of the CBR approach. A more detailed examination of scalability challenges and practical applications would enhance the depth of discourse.

(12) An objective evaluation and analysis of the study's methods in comparison to other scholarly works is recommended, as well as an assessment of the feasibility and prospective applications of the research.

(13) In the conclusion, the paper should acknowledge its limitations, analyze existing experimental challenges, and propose directions for future research.

Round 2

Reviewer 1 Report

Comments and Suggestions for Authors

No more comments

Comments on the Quality of English Language

Minor editing of English language required